# Health systems impacts of the COVID-19 pandemic on malaria control program implementation and malaria burden in Benin: A mixed-method qualitative and mathematical modelling and study

**Amber Gigi Hoi**[1‡]*, **Ludovic K. N'Tcha**[2,3‡], **Claudia Duguay**[1], **Manfred Accrombessi**[2,4], **Bruno Akinro**[2], **Cindy Feng**[5], **Ronald Labonté**[1], **Natacha Protopopoff**[4], **Martin Akogbeto**[2], **Manisha A. Kulkarni**[1]

1 School of Epidemiology and Public Health, University of Ottawa, Ottawa, Canada, 2 Centre de Recherche Entomologique de Cotonou, Cotonou, Benin, 3 Laboratory of Applied Anthropology and Education for Sustainable Development, University of Abomey-Calavi, Abomey-Calavi, Benin, 4 London School of Hygiene & Tropical Medicine, London, United Kingdom, 5 Department of Community Health & Epidemiology, Dalhousie University, Halifax, Canada

‡ AGH and LKN contributed equally and are considered joint first authors on this work.
* amb.g.hoi@gmail.com

**Data Availability Statement:** This mixed-methods study consists of two parts: key informant

## Abstract

The COVID-19 pandemic has sent ripple effects across health systems and impacted the burden of many other diseases, such as malaria in sub-Saharan Africa. This study takes a mixed method approach to assess the impact of COVID-19 on malaria control programs in three rural communes in Benin. We conducted individual semi-structured interviews with key informants who play important roles in malaria control in Benin at three levels of the health system–national, health zone, and commune. Using a purposive sampling technique, informants were interviewed regarding their roles in malaria control, the impact of the pandemic on their activities, and the mitigation strategies adopted. Relevant themes were identified by content analysis. We then formulated an agent-based model of malaria epidemiology to assess the impacts of treatment disruption on malaria burden. The key informant interviews revealed that essential aspects of malaria control were upheld in Benin due to the close collaboration of public health practitioners and health care providers at all levels of the health system. There were some disruptions to case management services for malaria at the start of the pandemic due to the public avoiding health centers and a brief shortage of malaria treatment that may not be entirely attributable to the pandemic. Results from the agent-based model suggest that duration, severity, and timing of treatment disruption can impact malaria burden in a synergistic manner, though the effects are small given the relatively mild disruptions observed. This study highlights the importance of top-down leadership in health emergencies, as well as the critical role of community health workers in preventing negative health outcomes for their communities. We also showcased the integration of qualitative research and mathematical models–an underappreciated form of mixed

interviews and a mathematical model. Excerpts of the interview transcripts are included in the main text of the manuscript. Data used to parameterize the mathematical model are all sourced from the literature and are included and referenced in the main text.

**Funding:** This study was funded by a grant from the International Development Research Centre, Canada to MAK (Grant No. 109551-001). The funders had no role in study design, data collection and analysis, decision to publish, or preparation of the manuscript.

**Competing interests:** The authors have declared that no competing interests exist.

methods research that offer immense value in the continued evaluation of rapidly evolving health emergencies.

## Introduction

The COVID-19 pandemic is having devastating impacts on the health of populations and health systems worldwide, with more than 760 million confirmed cases and 6.9 million deaths between December 2019 and June 2023 [1]. In addition to directly causing mortality and morbidity, the pandemic also has collateral impacts on the control and care of many other diseases and health conditions [2–6]. These unmet needs result from human and pharmaceutical resources being reallocated towards COVID-19 control and treatment, hospitals and other facilities being inundated with COVID-19 patients thereby limiting access to other services, and people avoiding health facilities due to lockdowns or fear of contracting COVID-19 [7]. The ripple effect of COVID-19 on health was expected to be most severe in regions such as sub-Saharan Africa where health systems are already precarious and the population are at risk of many other diseases [3, 7], including malaria [8, 9].

Malaria is endemic in many sub-Saharan African countries, where concerted disease control campaigns have resulted in the steady decrease in disease burden over the past two decades [10]. Established malaria control programs are highly effective: long-lasting insecticidal nets (LLINs) are the dominant prevention strategy [10], while prompt diagnosis and treatment with artemisinin-based combination therapies within 24 hours of illness onset is the key to effective case management [11]. However, the ultimate success of these interventions depends on strong leadership and coordination across all levels of the health system to ensure high coverage and uptake by individuals and households, especially in endemic rural areas. When the pandemic occurred, there were concerns that national LLIN campaigns would be disrupted, and avoidance of health facilities would cause delays in malaria treatment as well as low replacement rates of LLINs, which are distributed to end users through antenatal care and routine immunization visits [9, 12]. These disruptions, if realized, have the potential to jeopardize the progress towards malaria elimination made in the past two decades [9, 12].

Mathematical models were instrumental in illuminating the potential impacts of COVID-19 on malaria burden in sub-Saharan Africa in the early days of the pandemic by allowing researchers to assess the myriad of possible outcomes quantitatively and systematically [13–15]. Several models were analyzed for this purpose, and though their precise formulations differed, they were all based on the first principle that the pandemic may overburden health systems to a point that malaria control and treatment services could no longer be sustained [13–15]. In general, the impact of COVID-19 on malaria burden was found to depend on the duration and severity of service disruptions, and when the disruptions occurred relative to the onset of the malaria season and LLIN distribution campaigns [13, 15]. In the worst-case scenario where LLIN campaigns and treatment availability were both disrupted, it was predicted that malaria burden in 2020 could double that of 2019 levels [13, 15].

While mathematical models are undoubtedly powerful tools in pandemic management, many existing COVID-19-malaria models are based on speculations made at the beginning of the pandemic [13–15] and are thus due for updates. For instance, LLIN distribution campaigns scheduled in 2020 were assumed to suffer major setbacks [13, 15], however, most of these campaigns managed to be completed with only slight delays [16, 17]. Malaria diagnosis and treatment services, on the other hand, appear to have suffered widespread disruptions in delivery and a decline in access [2, 18]. Decreases in inpatient admission and outpatient consultations have been reported across sub-Saharan Africa since the onset of COVID-19, though the form

and magnitude of impact varied between countries and across different levels of health system within a country [2, 18, 19]. One study in Rwanda found a decrease in outpatient consultations at health facilities but visitations with community health workers increased [19]. A variety of reasons have been reported for the drop in malaria service levels in sub-Saharan Africa, including healthcare workers being redeployed towards COVID-19 care [20], difficulty accessing healthcare facilities due to closure or movement restrictions [20], and also the general public avoiding health facilities out of fear of contracting COVID-19 [21].

Benin, one of the West African countries most affected by COVID-19 with more than 280,000 confirmed cases and 160 cumulative deaths between Jan 3, 2020 to June 14, 2023 [1], has been at the forefront of the COVID-19 pandemic response. It was the first sub-Saharan country to pivot its national LLIN distribution campaign from a fixed-point distribution model to a digitized, door-to-door delivery model [16, 22]. The campaign was carried out between March 17–23, 2020, the same week the first case of COVID-19 was reported in Benin [1]. According to national monitoring center records, LLINs were delivered to ~95% of households nationwide, with coverage ranging from 90.59–96.15% across health zones [16]. Benin also implemented an observational laboratory-based assessment of SARS-CoV-2 molecular diagnostics and surveillance [23]. These programs and initiatives at the national level all aimed to alleviate the negative impacts of COVID-19 on malaria control, but it is unclear how these efforts trickled down to members of the community and, ultimately, how malaria burden was affected at that level.

Local and up-to-date knowledge is essential to identify high-risk populations and to inform effective and equitable policies and interventions to prevent a massive resurgence in malaria-related cases and deaths while ensuring effective pandemic responses [24]. This study employs a mixed methods approach to assess and evaluate the impact of the COVID-19 pandemic on malaria intervention programs in three rural districts in Benin. We had two interrelated objectives. First, we employed qualitative interviews to explore the nuanced impact of COVID-19 on malaria prevention and control at various levels of the health systems. Then, we used a mathematical model to simulate various scenarios of malaria intervention disruption due to COVID-19. Insights from the interviews allowed us to hone in on the likely scenarios and estimate the impacts of COVID-19 on malaria burden at the community level.

## Materials and methods

### Study area

The study site was located in the communes of Covè, Zagnanado, and Ouinhi, in the Zou health zone of central Benin, 154 km north of Cotonou, the economic capital of the country (Fig 1). This area consists of 123 villages with approximately 40,000 households and a population of 216,289. The main economic activities of the population are agriculture, fishing, hunting and trade. There are two rainy seasons, from April to July and from October to November. Malaria is highly endemic in this area and transmission is perennial due to floodplains with large pockets of stagnant water providing a favorable environment for mosquitoes to breed year round [17]. The dominant means of vector control in the Zou department are the distribution of LLINs as part of regular universal coverage campaigns (most recently in 2020 [16]) and during routine activities to pregnant women during prenatal care and to children under 5 years of age during expanded vaccination programs [25].

The malaria control program in Benin is administered through three levels of the health system: the national level, the health zone (or department) level, and the communal level (Fig 1). The national and health zone levels are primarily involved in policy development, strategy definition, information dissemination, staff training, stock control, and supervision of

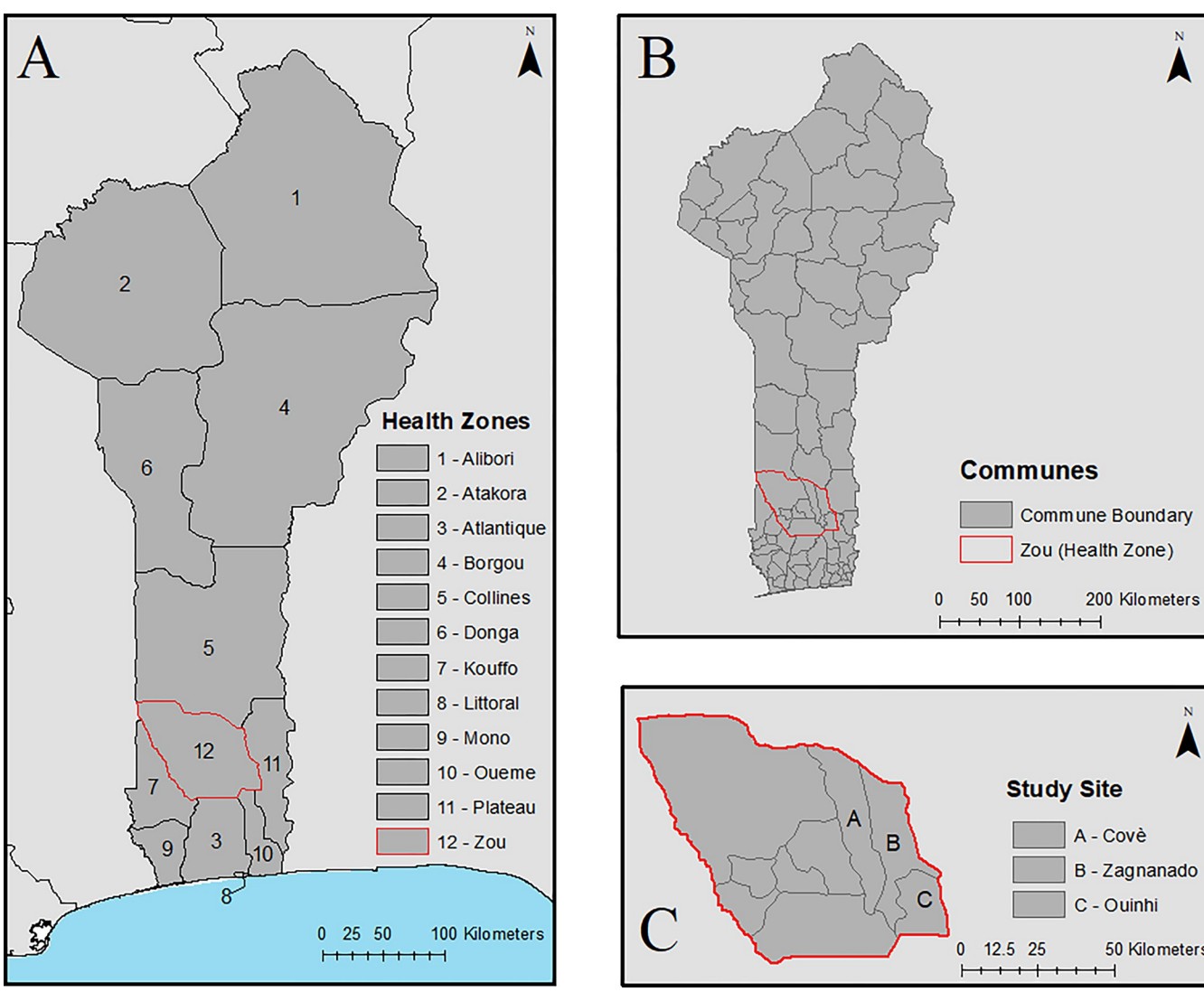

**Fig 1.** Maps depicting **(A) health zones in Benin, (B) communes within**, and the **(C) three communes (Covè, Ouinhi, and Zagnanando)** within the Zou health zone where this study took place. Map content was produced with Esri ArcGIS software using data provided by GADM [28] and Natural Earth [29].

implementation. Practical malaria prevention and treatment activities are carried out by health workers at the communal level, as they are the ones to interact directly with the public.

This current study capitalized on the infrastructure of the New Nets Project (NNP) [26] to include additional qualitative research and mathematical modeling components to understand the potential impacts of COVID-19 on malaria control. The NNP is a malaria intervention trial carried out between 2019–2022. The purpose of this trial was to evaluate the effectiveness of two new LLIN interventions for malaria control through a cluster randomized controlled trial. According to pre-trial surveys in 2019, malaria prevalence (all ages) in the study area was 43.5% [27], and LLIN coverage was 82.7% [25].

## Key informant interviews (KIIs)

We conducted semi-structured interviews with key informants to detail their roles in malaria control and how their work has been impacted by the COVID-19 pandemic. Key informants

were defined as individuals who play roles in malaria control in Benin, and were selected using a purposive sampling technique based on their positions in the health system. The target groups were: authorities in the National Malaria Control Program (NMCP), departmental health authorities, commune doctors, nurses, and community health workers, and members of the community. A total of 20 key informants were invited (three at the national level, two at the health zone level, and 15 at the communal level) and everyone agreed to participate.

The interviews were conducted between May 2021 and June 2021 by LKN and BA using an interview guide developed specifically for use with key informants (see S1 Text for details). Key informants were interviewed on their roles in malaria prevention and/or treatment, the impact of the COVID-19 pandemic on malaria control activities, and the measures taken to overcome these challenges. All key informant interviews were conducted in Fon, a local language, recorded using a dictaphone (IC Recorder, ICD-PX470, 2018–08), and transcribed verbatim in Microsoft Word 2013. A local translator then translated the interviews from Fon to French. The audio transcripts were then coded using Nvivo software through an inductive process and major and minor themes were identified through content analysis and thematic categorization. Select quotes were further translated into English for inclusion in this manuscript and the French versions of these quotes can be found in S2 Text.

## Agent-based model of malaria epidemiology

We used an agent-based model (ABM) of malaria epidemiology to assess the impacts of intervention disruptions on malaria burden. An ABM is a simulation-based mathematical model that simulates a complex system (e.g., malaria transmission) by tracking the behavior and interactions of individual agents (e.g., mosquitoes and humans) and their states of interest (e.g., age and disease status) [30]. Individuals are subjected to actions, such as intervention disruption, to which they will react based on their unique set of characteristics, and their states are tracked and updated over time. In our study, we used the ABM to simulate the transmission of malaria and assess the impacts of different types of treatment disruption on malaria burden. We chose this ABM approach over the classic Ross-Macdonald malaria models [31, 32] because it offers greater flexibility in modelling heterogeneities (e.g., individual variation in age, immune status, and time to recovery) and stochastic processes (e.g., exposure to infected mosquito), which are important factors driving malaria transmission dynamics [30].

In this study, we made use of a previous-published ABM that is available as an open-source R package, "malariasimulation" [33]. The model formulation was described in detail by Griffin et al. [34]. In brief, this model simulates a finite population of a given size and age distribution, and tracks individuals and their states over time. Individuals are "born" into the simulation with a certain level of maternal immunity to malaria, which wanes over time. Throughout their lives, individuals are exposed to infectious mosquito bites and may develop clinical disease, the probability and severity of which depends on their age and immunity level. Diseased individuals may recover naturally or via treatment and will gain immunity after. The rate at which individuals are exposed to infectious bites is determined by the local mosquito density and infectivity, which are derived from an independent model of vector population dynamics based on rainfall.

We first constructed a baseline model which simulates the seasonal malaria epidemic and the LLIN campaign in 2020 (which achieved 95.1% coverage in the Zou health zone [16]), absent the pandemic. This model was parameterized using local epidemiological and ecological data obtained from pre-intervention estimates of the NNP trial in the study area [25, 27]. These include population size (approximately 200,000), level and seasonality of exposure to malaria, and vector population characteristics (i.e., species composition). We also included a

7-year burn-in period to calibrate the model "starting conditions" (i.e., January 1st, 2020) to the appropriate level of LLIN coverage (estimated to be 82.7% in 2019 [25]) and conditions following the LLIN campaigns in 2014 and 2017. Due to the stochastic nature of ABMs, we replicated the simulation 5 times and calculated the mean and range of all-ages malaria prevalence over time.

We then built upon the baseline model to simulate the impact of changes in malaria intervention coverage on malaria burden. Given that the national LLIN distribution campaign in 2020 mostly carried on with only a slight delay [16, 22], we focused our simulations on disruptions to malaria treatment (formulated as the joint probability of seeking and subsequently receiving treatment). We took a sensitivity analysis approach due to lack of precise knowledge of the nature and degree of treatment disruption in the study area, and simulated treatment disruptions of increasing severity (from 0% to 100% disruption) and duration (from 1 to 6 months). We also considered the possibility that treatment disruptions may not occur immediately at the beginning of the pandemic (e.g., when supply chain delays are buffered by residual inventory), and simulated disruptions occurring at different time points in the pandemic (0–3 month offset from the start of the pandemic, assumed to be March 1). The three factors (i.e., the severity, duration, and timing of treatment disruption) and their various levels were simulated in a factorial manner (6x6x4 factorial, 144 combinations total) and each simulation of a unique factor level combination was replicated five times. Insights from the KIIs will inform which of the 144 scenarios considered are considered likely. The primary outcome of interest of the model is the all-ages malaria prevalence in October 2020, and we calculated its mean and range from the replicate simulations. We further compared model-predicted malaria prevalence with malaria prevalence of the reference arm (pyrethroid-only LLIN) of the NNP study population, assessed by a cross-sectional survey in October, 2020 [26].

### Ethics statement

The project protocol was approved by the institutional review board of the University of Ottawa (Certificate H-07-20-5944) (Canada), the institutional review board of the London School of Hygiene and Tropical Medicine (LSHTM Ethics Ref: 22637) (United Kingdom), and the ethical review committee of the Benin National Ethics Committee for Health Research (N° 047/MS/DRFMT/CNERS/SA) (Benin). Study participants were read an informed consent form by data collectors. This form was then read and signed by each participant prior to commencing interviews. Additional information regarding the ethical, cultural, and scientific considerations specific to inclusivity in global research is included in S3 Text.

## Results

### The structure of the health system in Benin in relation to malaria control

At the national level, the National Directorate of Public Health (DNSP) and the National Malaria Control Program (NMCP) are jointly responsible for malaria control activities. The DNSP is responsible for the preparation and coordination of malaria control programs whereas the NMCP is responsible for the implementation of these programs. Through its vector control service, the NMCP distributes mosquito nets in a mass campaign every three years and on a routine basis to vulnerable groups such as pregnant women and children under the age of five. The NMCP is headed by a National Coordinator, who coordinates all activities related to malaria control, makes supervision visits in the field, and ensures the supply of malaria control resources.

Health zones are set up at the sub-national level to decentralize the health system, and are the most operational level that provides care for the population. Each health zone is headed by

a coordinating physician whose main responsibility is to implement national policies and strategies for malaria control through the training and supervision of agents (e.g., doctors and nurses) on the various procedures, program evaluation, and inventory control at dispatching depots.

Each health zone is segregated into communes, where the malaria control team consists of doctors, nurses, community health workers (those who have received specific health training and serve in health centers at the district and commune levels), and community relays (those who facilitate disease prevention activities, follow up on treatment, and distributes health information at the commune level). Doctors are responsible for diagnosing and caring for cases of malaria (simple and severe) and for raising patient awareness of malaria prevention measures during prenatal consultation (CPN) sessions and the expanded vaccination program (EPI) visits for infants and young children. The nurses (or heads of posts) are in charge of receiving sick people at the health center, taking care of malaria cases after diagnosis, and the review of supplies at health centers (e.g., common drugs and rapid diagnostic tests, RDTs). The community health workers intervene at the local level, i.e., within the communities in the villages or neighborhoods of the commune. The community health workers are responsible for raising awareness among the population in villages and neighborhoods through exchange sessions or educational talks, diagnosing malaria using RDTs (especially with children aged 0 to 5 years), and supplying treatment to those presenting with uncomplicated malaria. In the case of severe malaria, nurses are responsible for the referral of the patient to the health center level for appropriate treatment and will follow up on these cases.

## Insights from key informant interviews (KII) on the impact of COVID-19 on malaria prevention and case management

The ways in which COVID-19 impacted malaria control in Benin, and the corresponding mitigation mechanisms placed at the various levels of the health system are described in detail below and summarized in Table 1.

The National Directorate of Public Health (DNSP) employed several strategies to curb the spread of COVID-19 and to minimize its impact on overall health care delivery, including malaria control. They created independent centers specializing in managing COVID-19 (screening, sensitization and management of infected cases) in all communes in an effort to keep COVID-19 out of regular health centers. DNSP also led the effort in training staff on COVID-19 knowledge, including its mode of transmission, associated symptoms, and preventative measures (e.g., wearing masks, washing hands, and social distancing). Of note, teams of 12 in each village ("community brigades") were trained on COVID-19 prevention and were tasked with sensitizing their communities and ensuring the preventative measures in place were respected. All health workers were supplied with personal protective equipment by the DNSP. Regular meetings (e.g., extended management committees, exchanges in the doctors' collective, review and information-sharing meetings) allowed information to be disseminated rapidly between all levels and enabled front-line health workers to be informed of the latest decisions made at the national level. A standardized protocol was also established by the NMCP to ensure that the guidelines were understood and followed by the health workers. Information on COVID-19 and malaria control in the context of the pandemic was regularly broadcasted through various channels (e.g., television, radio, and social networks).

> "*All this to free the health centers and minimize the spread of the disease among the staff at our health office. It is to allow the continuation of the offer of care and curative care consultations.*" (Cotonou, national level)

**Table 1. Summary of key informant interview results, organized based on whether the effect (or response) was expected, whether the effect on malaria control was positive or negative, and the level of the health system at which the effect was observed.**

| | Expected | | Unexpected | |
| --- | --- | --- | --- | --- |
| | Positive | Negative | Positive | Negative |
| **National level (n = 3)** | 1. Training of health workers to respond to COVID-19 and continue to maintain treatment and care for other diseases, such as malaria, in a safe manner.<br>2. Supply of personal protective equipment contained spread of COVID-19.<br>3. Creation of new centers to separate COVID-19 and malaria care.<br>4. Effective monitoring of malaria control supplies using an existing platform created by the NMCP.<br>5. Creation of a centralized platform used at all levels of the health system to respond to Covid-19.<br>6. Broadcasting of information on the response to COVID-19 and malaria control. | 1. Limited human resources spread thin between health emergence and existing challenges.<br>2. Overloading existing staff with new tasks. | 1. LLIN distribution campaign shifted from fixed point distribution to door-to-door distribution.<br>2. Face-to-face meetings were shifted online which saved resources (e.g., time, venue rental, and commuting costs). | 1. Budget for the LLIN campaign increased.<br>2. Reintroduction of chloroquine for the treatment of COVID-19, but then being used for malaria instead.<br>3. Rumors about COVID-19 spreading in health centers resulted in the avoidance of these centers. |
| **Health zone level (n = 2)** | 1. Training of community brigades in each village to sensitize communities on COVID-19 preventative measures.<br>2. Participation in local radio programs to inform the population about the pandemic and malaria control. | 1. Staffing shortage of health centers reduced access.<br>2. Existing staff inundated with new tasks. | 1. Face-to-face meetings were shifted online which saved resources. | 1. Decrease in the number of people attending health centers due to fear of contracting COVID-19. |
| **Community level (n = 15)** | 1. Frequency and attendance at information sessions decreased, but information continues to flow through home visits and direct communication, which includes direct follow-up of malaria cases. | 1. Staffing shortage of health centers reduced access.<br>2. Existing staff inundated with new tasks.<br>3. Short interruption in the availability of malaria control supplies. | 1. Information disseminated via local communication channels (e.g., community brigades, fellow villagers) was regarded more highly compared to other channels (e.g., television, radio). | 1. Decrease in the number of people attending health centers due to fear of contracting COVID-19.<br>2. Confusion about the use of chloroquine as treatment for COVID-19 and malaria. |

Malaria prevention and care activities had to adapt to the pandemic context, specifically, to ensure activities could be safely carried out while adhering to social distancing guidelines in place to curb COVID-19 transmission. The digitized LLIN campaign was successful [16], though it resulted in a considerable increase in the budget for the campaign. Many large-group physical meetings shifted online at the national and health zone levels, whereas at the community level where face-to-face meetings were inevitable, events either ran at reduced capacities, or pivoted to different formats. For example, malaria prevention information sessions that used to gather a lot of people were reduced from two sessions per month to one session that gathered no more than 15 people. Some sessions were replaced entirely by home visits during which the community health workers conducted individual interviews to pass on health messages directly.

"*When we directly went to people to offer them bed-nets, it was better. This ensured that the distribution went to the beneficiaries rather than providing a whole batch of bed-nets to a central location which would then distribute them. Each family received their bed-nets directly, which avoided the slowdown.*" (Cotonou, national level)

Access to malaria care in health centers was affected by the COVID-19 pandemic in several ways. Firstly, the NMCP was unable to increase staffing in a timely manner to handle new tasks related to managing the pandemic. Instead, existing staff were given additional duties

despite their already overloaded schedules, reducing the time available for other malaria-related activities. This was compounded by the public avoiding health centers out of fear of contracting COVID-19, especially at the beginning of the pandemic. Despite efforts to disseminate information on COVID-19 epidemiology and the measures taken to prevent its spread at health centers, this information was not always communicated accurately and rumors about health centers being a source of COVID-19 infections spread. People's reluctance to go to health centers further contributed to the reduction of malaria cases being managed there. After some time, the few patients who still went to health centers witnessed how seriously health workers took COVID-19 preventative measures. Their testimony along with few COVID-19 cases in the area eventually gave the public the confidence to return to the community health centers. The reduction in attendance at community health centers was also mitigated by community health workers providing home visits to follow up on malaria cases.

"*We have taken the necessary measures to be able to treat our patients normally. If it is not the phobia and panic of the population, they should have attended the health centers. But the population says that the disease is present in the health centers. It is this "false" information that makes communities flee the health centers.*" (Covè, community level)

Major disruptions in the supply chain throughout the pandemic were prevented through active monitoring of malaria control supplies (e.g., drugs, rapid diagnostic tests) through a centralized digital platform and proactive procurement of supplies. Given that most supplies are not manufactured in Benin, international transportation disturbances led to speculations of delays in the procurement of supplies at the national level during the beginning of the pandemic. However, only one instance of supply disruption was reported at a community health center, which was not attributable to the pandemic but to delays in requesting supplies from the health zone dispatching depot. This caused minor disruptions in the availability of certain products, such as artemisinin-based combination therapy (ACT), paracetamol, and rapid diagnostic tests (RDT), at the health center that lasted approximately two weeks.

"*Initially, this disruption is not necessarily dependent on Covid-19 directly at our level, perhaps at the national level. Since the products are not manufactured here in Benin, they are always imported*" (Covè, health zone level).

An unexpected indirect impact of the pandemic on malaria was the confusion surrounding the use of chloroquine. Chloroquine was previously authorized as a treatment for malaria in Benin, but was removed from the market in 2004 when drug resistance emerged and the drug ceased to be effective against malaria [35]. It was reintroduced during the pandemic due to disinformation on its treatment properties against COVID-19. The return of chloroquine to the pharmacies led people to request this drug out of familiarity, especially when ACTs were not available.

"*We had to go to chloroquine for some malaria treatments during this period of disruption before the health center was supplied with RDTs and ACTs.*" (Ouinhi, community level)

## Assessing the impact of COVID-19 on malaria burden using a mathematical model

We used an agent-based model (ABM) to simulate and assess the impact of COVID-19 disruptions on malaria burden. We calibrated our baseline model with a starting annual

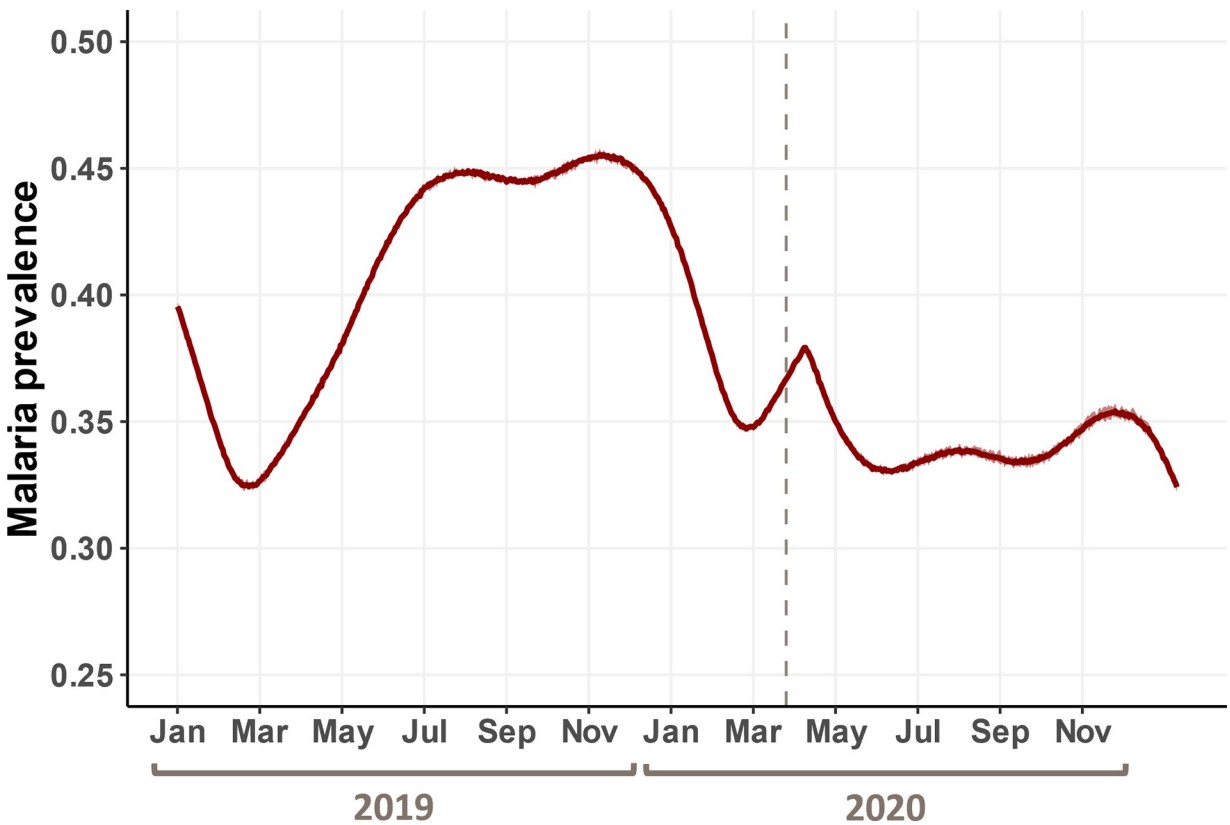

**Fig 2. Time series of malaria prevalence (all ages) in the baseline model, 2019–2020.** Solid dark-red line represents the mean malaria prevalence of the simulated replicates, and the light red band around it captures the range of simulated values. Dashed gray line marks the digitized LLIN distribution campaign (March 17–23, 2020) as well as the first reported case of COVID-19 in Benin (March 16, 2020).

entomological inoculation rate (EIR) of 150 to obtain a point malaria prevalence of 41.3% (in June 2019), similar to that observed in the reference population (43.5%) [27] (Fig 2). The baseline model was also able to effectively capture realistic malaria epidemiological dynamics in Benin, with the transmission season beginning around March, and prevalence remaining high for the remainder of the year.

Our simulations showed that the duration, severity, and timing of treatment disruption all influence malaria prevalence (Figs 3–4). As treatment disruption increased in severity, malaria prevalence increased correspondingly but maintained the same seasonal dynamics throughout the year (Fig 3A). In contrast, the duration and timing of disruption both impacted the seasonal dynamics of malaria qualitatively. COVID-19 reached Benin at the beginning of its malaria transmission season, i.e., when mosquito abundance began to increase. When treatment disruptions were prolonged, a new peak in prevalence emerged due to the accumulation of untreated cases (Fig 3B), whereas when the onset of disruption is offset from the start of the pandemic, peak malaria prevalence was pushed later and to higher levels due to treatment being removed when the entomological risk was increasing (Fig 3C). The effect of disruption duration and severity on malaria prevalence was synergistic such that the effect of prolonged disruption becomes more apparent as severity increases (Fig 4). This interaction is slightly exacerbated when there is some offset between the onset of treatment disruption and the start of the pandemic (Fig 4A vs. 4B–4D), though this effect appeared to be subtle. Point malaria prevalence (on Oct 1, 2020) was estimated to differ by ~5% when comparing the best- and

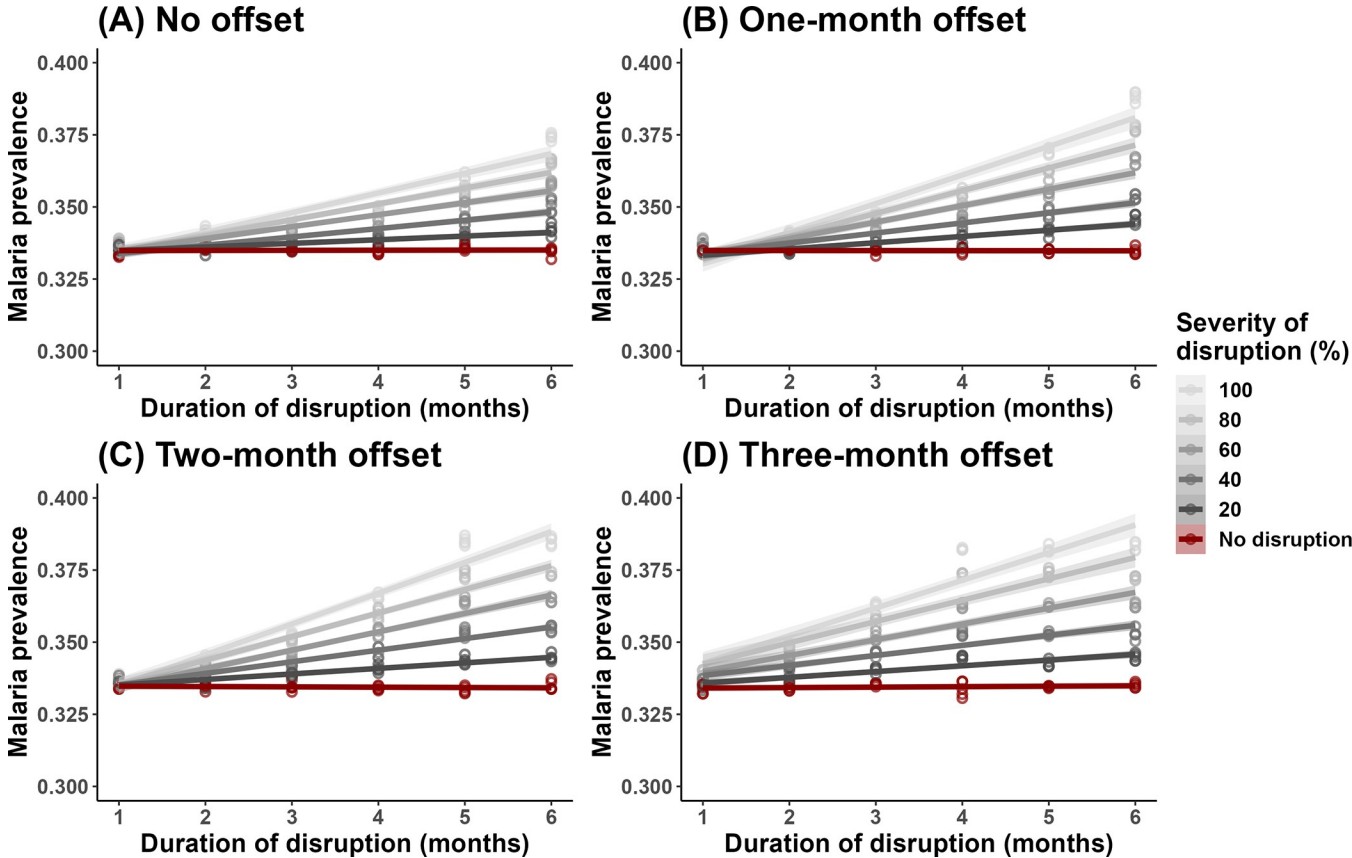

**Fig 3.** Simulated malaria dynamics for the year 2020 with **A) varying severity of treatment disruption**, where disruption began on March 1 and lasted 6 months, **B) varying duration of treatment disruption**, where there was 100% disruption that began on March 1, and **C) varying timing of treatment disruption onset**, where treatment was 100% disrupted for 6 months. Solid color lines represent the mean malaria prevalence of the simulated replicates, and the light color bands around them capture the range of simulated values. Dashed gray line indicates Oct 1, 2020, and the red cross indicates the observed malaria prevalence in the NPP study reference group (28%) [26].

worst-case scenarios (i.e., ~33% for no disruption vs. ~38% for total disruption for 6 months, regardless of offset). Even in the best-case scenario of no treatment disruption, the model estimated malaria prevalence was higher than that obtained from the cross-sectional survey (33.4% vs 28% [26]).

## Discussion

The COVID-19 pandemic swept across an ill-prepared world in March 2020. Little was known about the pathophysiology and epidemiology of the virus, and there was a lot of uncertainty surrounding the rapidly evolving situation. Mathematical models proved to be an indispensable tool in those early days for forecasting the trajectory of the pandemic (e.g., [36]), as well as evaluating and guiding responses towards COVID-19 (e.g., [37–39]) and other health programs affected (e.g., [14]), malaria control among them [13, 15]. As the pandemic persists, these models need to be refined with newly emerged data to better strategize as we move forward [40]. Ideally, we would have longitudinal quantitative data with records starting before the onset of the pandemic, on both the exposure and outcome of interest for model calibration and validation. Unfortunately, such data is rarely available, for example, due to missing pre-pandemic baseline comparators. Critical aspects of pandemic management, such as

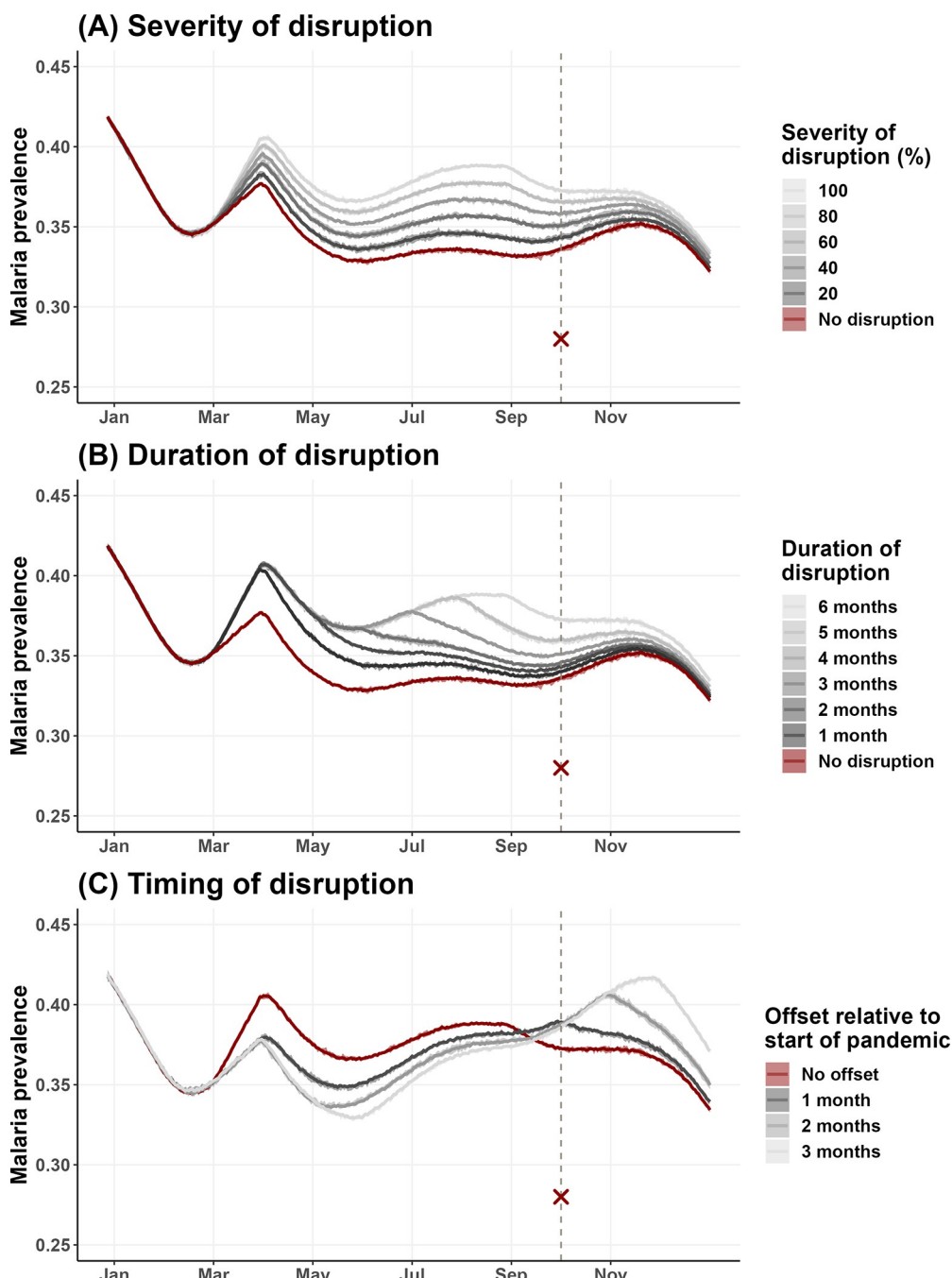

**Fig 4. The effect of treatment disruption on model-simulated point malaria prevalence (all ages) on Oct 1, 2020.**
Several aspects of treatment disruptions were simulated in a factorial manner, including duration of disruption (1–6 months, along x-axes), extent of disruption (0–100%, colored lines), and onset of disruption relative to the beginning of the pandemic (0–3 month offset from March 1, 2020, panels A-D). Points represent individual simulations, solid color lines represent the regression line of malaria prevalence on duration of treatment disruption grouped by severity of disruption, and light color bands represent the 95% confidence interval of the regression line.

collaboration between government agencies and societal engagement, are also often difficult to quantify [41]. The contribution of qualitative data to pandemic management should therefore not be overlooked. Not only can qualitative data provide important insight into the decision

making and policy implementation processes, they can also guide model design and aid in narrowing down the parameter space (i.e., all possible values for all of the parameters included in the model, and combinations thereof) that is needed.

In this study, we used a qualitative approach (key informant interviews, KII) to assess the impact of the COVID-19 pandemic on malaria intervention programs in three rural districts in Benin and then used a mathematical model (agent-based model, ABM) to evaluate the effects of these impacts on malaria burden. We found that essential malaria control and treatment services provided by the three levels of the health system (i.e., national, health zone, and community) were differently affected by the pandemic, but the pandemic's overall impact on malaria burden appeared to be small. One of the most significant achievements led by Benin's National Malaria Control Program (NMCP) was the successful completion of the LLIN distribution campaign [16]. The new door-to-door distribution model ensured effective delivery of LLIN and increased coverage, though it also resulted in an increase in budget. Regarding treatment disruption, two major themes emerged from the KII that were in line with how this factor was formulated in the ABM: treatment seeking behavior, and treatment availability. Key informants reported a decrease in attendance at both zonal and community health centers due to a combination of avoidance behavior as well as staffing shortages limiting access. The negative consequences of this disruption were alleviated in part by community health workers and community relays providing home visits and follow-up of malaria cases. Malaria control supplies were maintained effectively by the NMCP at the national level, and even though there was one report of a short disruption in the availability of supplies at the beginning of the pandemic, it was most likely indirect rather than due to any pandemic influence.

Our qualitative results suggest only a mild and short treatment disruption at the beginning of the pandemic. Under this scenario, the ABM estimates little detectable change in malaria prevalence in our study population (Fig 4A and 4B, comparing red and dark gray lines). More broadly, the sensitivity analysis of the ABM showed that even in the worst case scenario of very prolonged (6 months) and severe treatment disruption (100% disruption), peak malaria prevalence increased only by ~5% (Fig 4, regardless of offset). We hypothesize that the small impact of treatment disruption on malaria burden was due to the very high coverage of bednets (95% in the model [16]) being overwhelmingly protective. The ABM simulations also demonstrated that certain treatment disruption scenarios can change malaria dynamics qualitatively, where the timing of the upswing and peak is pushed back and the transmission season prolongs, suggesting the need to remain diligent outside of the typical transmission season.

This study is uniquely positioned in that it was nested within an ongoing LLIN trial (New Nets Project, NNP [26]) where the study population was actively monitored. Comparing ABM model results to a cross-sectional malaria survey of the NNP reference group (in October 2020, Fig 3), we find that the observed malaria prevalence is lower than even the baseline simulation scenario of no treatment disruption (28% vs. 33.4%; [26] & Fig 3). This was undoubtedly a good outcome for the local community, however, perplexing in the implication that the pandemic appeared to have a protective effect on malaria burden. A potential explanation for this discrepancy is that, due to the ongoing NNP trial, sufficient health care workers were serving in the community, with those enrolled in the study cohort receiving regular visits by health care workers who tested and treated those who presented with malaria symptoms [26]. As such, malaria treatment was probably not disrupted, and may even have been strengthened, meaning that malaria prevalence in the NNP reference group would not be representative of that in the general population. Another possibility could be that the model was not calibrated accurately because we lacked estimates for treatment coverage levels pre-pandemic. In the current model, pre-pandemic treatment coverage was set to 100%, which is unrealistic but can still serve as a useful point of reference for pandemic levels and comparison between simulated

scenarios. Model simulations also revealed that the cross-sectional survey was likely not carried out when the impact of disruption was predicted to be the most pronounced (September 2020 for the effects of severity and duration of the disruption, and December 2020 for the effect of timing of disruption; Fig 3), which could contribute to the difference between observed and estimated malaria prevalence.

How the malaria season in Benin transpired during the early stages of COVID-19 has important implications for future pandemic management. Our study results highlight the importance of swift top-down leadership in maintaining essential health services, but more importantly, in setting up local programs that mobilize community members to disseminate information and educate each other. It was clear that community health workers played an important role in alleviating the impact of staffing shortages and health facility avoidance behavior at our study sites. It is also known from the literature that community-led (not just community-based) programs that take into account local socio-economic and cultural factors are more likely to secure participation and ownership from the targeted communities [42]. The trust between community health workers and the public they serve is also invaluable–in addition to information being accurate, it is equally important to consider who is delivering it such that it can be fully received. Strengthening community and primary health care will be crucial in preparing for future health emergencies, as well as improving the health of communities in general [43]. Follow-up studies that evaluate the long-term impacts of COVID-19 on the Benin health system and malaria control programs will further shed light on the efficacy of these early response strategies.

## Conclusion

More than three years have passed since COVID-19 was declared a pandemic, and it is far from over. This is especially poignant due to unequal access to the vaccine, where 73% of residents in high-income countries have received at least one dose of the vaccine, compared with just 32% in low-income countries [44, 45]. Calls to action to end COVID-19 and to prepare for the next pandemic are abundant, all converging on the themes of building long-term health system capacity, building trust, and reducing long-standing inequities and injustices [46–49]. All these will benefit all people and all diseases, both in times of emergency and otherwise. The world is learning a lot about effective pandemic response, and rapidly developing new tools to help us dissect the complexities and uncertainties within. With this study, we hope to make a positive contribution towards this knowledge and toolkit by demonstrating the utility of mixed-methods research, specifically, how mathematical models and qualitative surveys can complement each other.

## Supporting information

**S1 Text. Key informant interview guide.**
(DOCX)

**S2 Text. Quotes related to the impact of COVID-19 on malaria control in French, presented in order of inclusion in the main text.**
(DOCX)

**S3 Text. Inclusivity in global research questionnaire.**
(DOCX)

## Acknowledgments

We thank all study participants and leaders of the Benin government for agreeing to collaborate on this study. We also thank Prof. Franklin Mosha for his contribution to the early

conceptualization of this study, and Dr. Ellie Sherrard-Smith and Mr. Giovanni Charles for their guidance with the agent-based model of malaria and "malariasimulation" R package.

## Author Contributions

**Conceptualization:** Manfred Accrombessi, Cindy Feng, Ronald Labonté, Natacha Protopopoff, Martin Akogbeto, Manisha A. Kulkarni.

**Data curation:** Ludovic K. N'Tcha, Bruno Akinro.

**Formal analysis:** Amber Gigi Hoi, Ludovic K. N'Tcha, Claudia Duguay.

**Methodology:** Amber Gigi Hoi, Ludovic K. N'Tcha, Claudia Duguay, Manisha A. Kulkarni.

**Software:** Amber Gigi Hoi, Ludovic K. N'Tcha, Claudia Duguay.

**Supervision:** Manfred Accrombessi, Martin Akogbeto, Manisha A. Kulkarni.

**Visualization:** Amber Gigi Hoi.

**Writing – original draft:** Amber Gigi Hoi, Ludovic K. N'Tcha.

**Writing – review & editing:** Amber Gigi Hoi, Ludovic K. N'Tcha, Claudia Duguay, Manfred Accrombessi, Bruno Akinro, Cindy Feng, Ronald Labonté, Natacha Protopopoff, Martin Akogbeto, Manisha A. Kulkarni.

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
