## [Decision Letter · Decision Letter 0]

25 Oct 2023

PGPH-D-23-01173

Health systems impacts of the COVID-19 pandemic on malaria control program implementation and malaria burden in Benin: a mixed-method qualitative and mathematical modelling and study

Dear Dr. Hoi,

Thank you for submitting your manuscript to PLOS Global Public Health. After careful consideration, we feel that it has merit but does not fully meet PLOS Global Public Health’s publication criteria as it currently stands. Therefore, we invite you to submit a revised version of the manuscript that addresses the points raised during the review process.

We look forward to receiving your revised manuscript.

Kind regards,

Eunha Shim

Academic Editor

Journal Requirements:

2. "Please provide separate figure files in .tif or .eps format.

Additional Editor Comments (if provided):

Reviewers' comments:

Reviewer's Responses to Questions

**Comments to the Author**

1. Does this manuscript meet PLOS Global Public Health’s publication criteria? Is the manuscript technically sound, and do the data support the conclusions? The manuscript must describe methodologically and ethically rigorous research with conclusions that are appropriately drawn based on the data presented.

Reviewer #1: Yes

Reviewer #2: Yes

2. Has the statistical analysis been performed appropriately and rigorously?

Reviewer #1: I don't know

Reviewer #2: N/A

3. Have the authors made all data underlying the findings in their manuscript fully available (please refer to the Data Availability Statement at the start of the manuscript PDF file)?

Reviewer #1: Yes

Reviewer #2: Yes

4. Is the manuscript presented in an intelligible fashion and written in standard English?

Reviewer #1: Yes

Reviewer #2: Yes

5. Review Comments to the Author

Reviewer #1: The manuscript is well written and the finding support the conclusion.

I have some concerns that I am suggesting that may help authors to improve transparency.

Materials and methods are suggested to be reported in separate sections as per the STOBE guidelines.

The same goes for the results to slip into the separate sub-sections that support answering the research questions.

The author may find this paper interesting to cite as the possible impact of covid on mosquito-born disease is discussed in this article: https://doi.org/10.18332/popmed/145535

the impact of the covid on the health system and the nurses in the context of a developing country are evident in the following articles and the author may find them interesting to cite:

https://doi.org/10.1186/s43045-021-00103-x

https://doi.org/10.1016/j.heliyon.2023.e13162

https://doi.org/10.1111/inr.12802

https://doi.org/10.1371/journal.pone.0274965

Reviewer #2: Appreciate the authors for this study to find out the impact of Covid19 on malaria control programs by using mixed methods in three different levels of health system. The authors have presented the expected and unexpected and also positive negative impacts at the various levels of health system and formulated an agent-based model of malaria epidemiology to assess the impacts of treatment disruption on malaria. The effort put in by he authors are laudable.

The limitation is that this study was done in the early stage of covid when the health system was unaware of many aspects also about resilience. Later on, lot of developments have happened regarding the disease Covid 19, the mitigation strategies, how to bring in resilience etc. So, this manuscript may appear as if not relevant for today’s situation. But this study could be an inspiration to have a further look at the same thing at a later date thereby understanding more about the impact of Covid 19 on Malaria control programs in a changed health system and the lesson could have far reaching importance like prevention of further pandemics. So authors can elaborate more on the limitations of the study and the scope for further studies related to this one .In addition more details to be given related to the integration of qualitative and quantitative methods in the context.

6. PLOS authors have the option to publish the peer review history of their article (what does this mean?). If published, this will include your full peer review and any attached files.

**Do you want your identity to be public for this peer review?** For information about this choice, including consent withdrawal, please see our Privacy Policy.

Reviewer #1: No

Reviewer #2: No

---

## [Editor Report · Decision Letter 1]

4 Jan 2024

Health systems impacts of the COVID-19 pandemic on malaria control program implementation and malaria burden in Benin: a mixed-method qualitative and mathematical modelling and study

PGPH-D-23-01173R1

Dear Ms Hoi,

We are pleased to inform you that your manuscript 'Health systems impacts of the COVID-19 pandemic on malaria control program implementation and malaria burden in Benin: a mixed-method qualitative and mathematical modelling and study' has been provisionally accepted for publication in PLOS Global Public Health.

Best regards,

Eunha Shim

Academic Editor